# Mathematical Modelling of Biosensing Platforms Applied for Environmental Monitoring

**Ahlem Teniou [1], Amina Rhouati [1,\*] and Jean-Louis Marty [2,\*]**

[1] Bioengineering Laboratory, Higher National School of Biotechnology, Constantine 25100, Algeria; teniouahlem97@gmail.com

[2] UFR Sciences, Universitéde Perpignan Via Domitia, 66860 Perpignan, France

\* Correspondence: a.rhouati@ensbiotech.edu.dz (A.R.); jlmarty@univ-perp.fr (J.-L.M.)

**Abstract:** In recent years, mathematical modelling has known an overwhelming integration in different scientific fields. In general, modelling is used to obtain new insights and achieve more quantitative and qualitative information about systems by programming language, manipulating matrices, creating algorithms and tracing functions and data. Researchers have been inspired by these techniques to explore several methods to solve many problems with high precision. In this direction, simulation and modelling have been employed for the development of sensitive and selective detection tools in different fields including environmental control. Emerging pollutants such as pesticides, heavy metals and pharmaceuticals are contaminating water resources, thus threatening wildlife. As a consequence, various biosensors using modelling have been reported in the literature for efficient environmental monitoring. In this review paper, the recent biosensors inspired by modelling and applied for environmental monitoring will be overviewed. Moreover, the level of success and the analytical performances of each modelling-biosensor will be discussed. Finally, current challenges in this field will be highlighted.

**Keywords:** modelling; MATLAB; biosensors; pollutants; environmental control

## 1. Introduction

During the last decade, Artificial Intelligence (AI) gained tremendous advances in different applications. It has been used to solve complicated, nonlinear and dynamic problems. Besides, it is considered as an alternative approach to conventional procedures, or as a component of integrated systems to perform modeling, prediction, simulation and optimization at high speed [1]. AI technologies principally refer to artificial neural network (ANN), genetic algorithm (GA) and expert system (ES) chemometric methods. These technologies have been applied to agriculture, climate, finance, engineering, environment, education, medicine, nanotechnology and various disciplines [2–4]. In general, AI can be considered as the ability of a machine to mimic functions that characterize human thought in order to be able to perform very complex tasks. According to Barr and Feigenbaum, AI is the part of computer science concerned with the design of intelligent computer systems, i.e., systems that exhibit the characteristics associated with intelligence in human behavior, understanding, language, learning, reasoning, solving problems and so on [5–7]. AI technology systems have been gradually improved and have emerged as a powerful and promising technique in developing reliable, low cost and rapid intelligent biosensors to treat a large number of data sets, beyond the invariant technique. Moreover, the integration of AI approaches with biosensors can fill the gap between data acquisition and analysis, which can lead toeffective and accurate decision making [8].

Environmental monitoring is one of the growing concerns in the world because of the hazardous effects of pollutants on human health and ecosystems [9, 10]. The massive

global contamination of atmosphere, water, and soil is mainly produced by industrial, wastewater, and domestic effluents through pesticides, heavy metals, pharmaceutical drugs and biotoxins which are considered as the most commonly observed environmental pollutants [11,12]. Unfortunately, most of these contaminants are non-biodegradable with high toxicity and a long half-life leading to bioaccumulation and increasing the risk facing living organisms [13,14]. Therefore, environmental monitoring plays a key role in preventing the dangerous effects of these contaminants. Traditionally, their assessment is based on laborious techniques including high-performance liquid chromatography (HPLC), gas chromatography (GC), capillary electrophoresis, mass spectroscopy, and thin layer chromatography [15].Such methods require a huge number of experiments which would be expensive, time consuming and lead to uncertain estimations [16,17]. AI technologies are able to overcome problems of conventional mechanisms since they allow complex mathematical formula computation with detailed information and without the loss of precision [5]. Many efforts have been thus devoted to the integration of AI in developing accurate biosensors for environmental pollutants. In addition, a large number of studies have confirmed that AI technologies are good assistants for environmental pollution control [18–21]. In view of the wide use of AI to monitor environments, this review will first introduce environmental pollutants and their hazardous effects. Then, we will present an overview of fundamental concepts of the recent AI technologies. Finally, the last part will be dedicated to discussing the different applications of AI methods in simulation, prediction, optimization, modeling and intelligent biosensing of pollutants.

## 2. Major Environmental Toxic Substances and Their Effects

Nowadays, the whole world suffers from toxic substances, mainly produced by human activities, the pharmaceutical industry, rapid industrialization, and unplanned urbanization [22], and the textile industry and its dye-containing wastewaters is considered as one of the major sources of pollution [23]. The wide use of chemicals may cause cancer, chronic diseases and alter reproductive systems, in addition to damaging developing brains even at low levels of exposure [24]. Natural and synthetic chemical substances harmfully affecting the environment, ecosystems and human health are called emerging pollutants (EP). Nguyen et al. classified environmental toxic agents into four categories: toxins, pesticides, environmental polluting hormones and persistent organic toxic chemicals (POTC), pharmaceuticals and personal care products (PPCPs) [25].

### 2.1. Pesticides

Pesticides are chemical substances that destroy weeds, pests, diseases, insects, etc., by disturbing the target physiological activities, causing dysfunction, and vitality decline [26]. They are mainly used to regulate damages in order to maintain a high product quality, ensure a high profit, minimize loses, and even to enhance the nutritional value of food. However, it has been shown that 99.7% of pesticides go into the environment contributing to pollution and only 0.3% will hit the target [27]. Because they have been developed to destroy certain organisms, pesticides are highly toxic and harmful not only for human beings, but they constitute a principal contamination source for the environment, ecosystems, wildlife, aquatic systems and terrestrial species [28–30].

Some pesticides have a large range of targets, others are specific, and their names depend on the targets for which they are synthesized (insecticides, fungicides, herbicides, etc.) [26,31]. Classification of pesticides is mainly based on their chemical nature (organochlorines, organophosphates); application requirement (agriculture, public health, domestic); and target organism or targeted use (insecticide, herbicide, fungicide, etc.) [32].

The highly extended use of pesticides has increased with the appearance of new targets (new pests, new weeds). In parallel, lipophilicity, bioaccumulation, and the long half-life of pesticides have been observed to lead to increased contamination of

environmental factors (water, soil, air). This has also altered the food chain and ecosystem balance causing hazardous health issues [32–34].Serious health risks have been detected as a result of high exposure to pesticides mainly through residues in food and drinking water. Many cause hypertension and cardiovascular disorders [35], others are cancerogenic, neurotoxic, and act as endocrine disrupting chemicals [36].

## 2.2. Heavy Metals (HMs)

HMs are a group of metals and metalloids characterized by a high density (higher than 4000 kg/m$^3$) and a high toxicity even at low concentrations [37,38]. This class of emerging pollutants comprises arsenic (As), lead (Pb), mercury (Hg), and cadmium (Cd). Copper (Cu), selenium (Se) and zinc (Zn) are also trace elements that are considered as heavy metals [39]. Most of heavy metals are produced by industrial processing, mining, automobiles, pharmaceuticals, electroplating, organic chemicals, and other industrial wastewater [40,41]. HMs cause huge damage to human life, plant metabolism, ecosystems, aquatic systems, and the environment largely speaking [42].

Certain HVs are essential for our body at low concentrations, but they become harmful by exceeding the permissible limit. Respiration, ingestion, and skin are the principal body's entry ways for HVs [39]. Heavy metals interfere with the body's systems by binding to specific enzyme/proteins and forming 'free radicals' which repress the access of nutritional minerals by entering into competition with them. Thus, HMs can alter some cellular functions, metabolism, and other substances that are essential to maintain the organism's balance [43]. Chronic exposure to heavy metals induces mutagenicity, carcinogenicity and immunosuppression. Moreover, they may damage kidneys and liver and alter the levels of different biomarkers and hormones [44]. For instance, high levels of Pb affect hemoglobin synthesis, kidneys, and reproductive and nervous systems [45,46]. For living organisms, mercury inhalation is the most dangerous and toxic means of exposure, and can cause severe disease especially to neural and renal systems [47].

The widespread presence of heavy metals in the environment leads to unnatural growth change in plants, causing an acute problem of pollution in farming soil and quality of production [38]. It has been demonstrated that oxidative stress and reactive oxygen species are mainly produced by the high concentrations of Cu in plants [48,49]. A significant rate of Pb in soil has also been associated with the altered morphology of different plant species [50]. Moreover, a continuous high exposition of plants to Cr influences the photosynthesis process by affecting carbon dioxide fixation, the activities of enzymes, photophosphorylation and electron transport [51,52]. Furthermore, the liberation of heavy metals into aquatic systems may result in various physical, chemical and biological processes [53]. For instance, changes in physical condition may include water pH, organic content of substrate and size of particles in water, thus affecting plants by reducing species composition, diversity, and density [54–56].

## 2.3. Pharmaceuticals

In the last few decades, wastewater, drinking water, and superficial water have been found to be highly contaminated by pharmaceuticals produced by households and hospitals which are the most important sources of these substances' emission [57]. Hormones, lipid regulators, pain killers, antibiotics, anti-cancer drugs, and other active substances have been detected in different environmental situations [58]. Human excretions are the major sources of these drugs; after their consumption, pharmaceuticals are excreted unaltered (unchanged) and/or as metabolites [59,60].

Most active substances (such as quinolone, sulfonamide) have a low biodegradability and cause hazardous effects on humans and the environment [61,62]. For instance, cytostatic agents and immunosuppressive drugs are mutagenic, cancerogenic and embryotoxic. Antibiotics and disinfectants are also dangerous because of their toxic bacteria and risk of fostering resistance. Due to their important biological activity and large use in agriculture, livestock and farming, antibiotics are widely contaminating soils,

seas, and drinking water. Their negative effects make them one of the major contaminants for the environment, water, and food [63]. Illicit drugs (medical substances are used only for a medical aim; all non-medical uses of these drugs are forbidden by law) [64] are classified as the latest group of emerging pollutants that mainly affect water and the environment [65,66]. For example, cocaine, morphine, amphetamine, and MDMA (3,4-methylenedioxymethamphetamine) have an important pharmacological activity. Their dangerous impact on aquatic organisms and human health cannot be neglected [67]. These drugs are released in wastewater as unaltered drugs or active metabolites produced by illegal laboratories or illicit consumption [68,69].Even at low concentrations, illicit drugs can form a toxic complex with other organic compounds or therapeutic drugs through pharmacological interaction that may cause hazardous effects for organisms [70,71].

*2.4. Biotoxins*

Biotoxins can be defined as toxic substances or products generated by plants, animals and microorganisms [72]. They are essentially produced by harmful bacteria, alga bloom or fungi. Biotoxins are responsible for several disorders threatening humans and wildlife through their carcinogenicity, mutagenicity, and toxicity [73–75]. The large propagation of biotoxins globally threatens domestic and international trade. The FAO, and EU,US have thus increased biotoxin limits to a strict maximum [76]. Mycotoxins, algal, bacterial, and plant toxins are the major group affecting the environment [77]. Among them, anatoxin-A (ATX), cylindrospermopsin (CYN), and microcystins (MCs) are the most common toxin groups found in freshwater. We can cite also brevotoxin (BTX), okadaic acid (OA), palytoxin (PTX), saxitoxin (STX) and others, which are principally classified as marine toxins [78]. In addition, aflatoxins (AFs), fumonisins (FBs), ochratoxin A (OTA), trichothecene, and deoxynivalenol (DON) secreted by *Aspergillus* and *Penicillium* are highly toxic mycotoxins found in food and plants [79,80].

## 3. Advanced Analysis Techniques Based on Artificial Intelligence (AI)

The fifth-generation modeling system combines AI technology and computational hydrodynamics to assist non-experimented users [81–83]. Numerical modeling is described as a process aiming to transform knowledge (physical, biological, chemical processes, etc.) into digital formats, to simulate behaviors and translate the results into comprehensible formats [84]. AI mimics human intelligence on a machine in order to improve its efficiency to solve problems using knowledge [85]. Nowadays, AI represents a promising alternative to conventional techniques for complex problems in numerous fields. Due to its symbolic reasoning, flexibility and explanation capabilities, AI is qualified to process nonlinear problems. It is applied for identification, optimization, prediction and forecasting [2]. AI comprises several technologies, such as Expert Systems (ESs), Artificial Neural Networks (ANNs), Genetic Algorithms (GAs), Fuzzy Logic (FL), Problem Solving and Planning (PSP), Non-Monotonic Reasoning (NMR), Logic Programming (LP) and others[86]. Some of these types are discussed in the present review.

*3.1. Expert Systems (ESs)*

Also called Knowledge Based Systems (KBS), this consists of a computer program that uses a knowledge base of human experts in solving problems [87]. This knowledge allows experts to specify rules simulating the process of thinking by providing simple plans to draw conclusions and solve problems. KBS can also be used for data interpretation and selecting the right decision from a list of alternatives.

ES has three principal parts: a user interface to communicate with users; an inference machine acts as a decision maker; and a knowledge basecollected from books, magazines, and experts [88]. KBS has a wide range of applications since it is able to achieve a high

level of performance in comparison to that of human experts [83]. For instance, ES can be used to enhance medical diagnosis systems' quality, particularly in term of precision [89,90]. Figure 1 displays the principal components of the expert system.

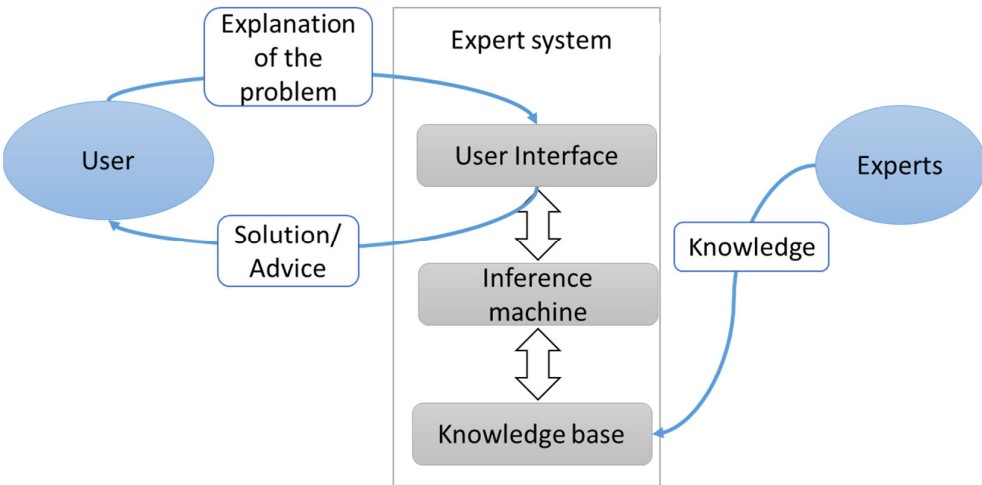

**Figure 1.** Architecture of expert system. Adapted from Maylawati et al. [91], and Salman and Abu-Naser [88].

### 3.2. Genetic Algorithms(GAs)

GAs are one of the evolutionary algorithms using computational models of natural evolutionary processes in developed computer-based problem-solving systems [92]. This system imitates the process of natural genetics and biological mechanisms. It can be used to optimize an objective function, comprehend model prediction and behavior, and determine patterns and relationships, in addition to driving certain phenomena [93].

### 3.3. Chemometrics

Chemometric approaches are widely used in analytical chemistry, in particular environmental studies, showing the potency of data processing techniques in this field. Quantitative chemical analysis, environmental quality assessment monitoring, modeling, and prediction of toxicological effects are the major areas of interest in chemometric environmental studies [94,95]. The high potential of chemometrics resides in obtaining reasonable analytical results from poor quality of data: low resolution, strong signal overlapping, high level of noise, etc. [96]. Moreover, chemometric techniques improve sensitivity and selectivity, and lower the detection limits of analytical tools. Data description and visualization, detection of hidden relations between analytical signals and sample parameters, discrimination, classification, regression, and prediction are also performed using chemometric tools [97]. Furthermore, chemometrics suggest solutions for complex problems by providing pattern recognition of chemical profiles of environmental or food samples, frequently containing a number of markers to identify [98].

There are many chemometric techniques, such as: Principal Component Analysis (PCA), Partial least squares regression (PLSR), Cluster Analysis (CA), Linear Discriminant Analysis (LDA), Random Forest (RF), etc. These techniques are classified as supervised or unsupervised tools [99].

### 3.3.1. Unsupervised Methods

These are widely used in exploratory analysis of the global structure of a dataset, and in finding trends and patterns within the dataset. In terms of classification or discrimination of samples, these methods should not be misused. Principal Component

Analysis (PCA), Hierarchical Cluster Analysis (HCA), and Partial Least Squares Regression (PLSR) are the most frequently used [100].

Principal Component Analysis

PCA is considered as the most powerful and popular chemometric technique, and is the basis of several other chemometric methods. This approach is a multivariate statistical method usually used in exploratory data analysis [101]. In general, all the data intended for processing are compiled in a matrix form (called X). Each row of this latter matrix contains raw data (variables) in order to describe each studied sample. The PCA is the decomposition of the X matrix with n rows (samples) and p columns (variables) into the product of scores matrix T and transposed loadings matrix P plus residuals matrix E (Figure 2). The scores are the position of the samples in the space of the principal components (PCs) while the loadings are the contributions of the original variables to the PCs [102]. In order to simplify structures and illustrate a large amount of data, PCA is used to calculate a smaller number of possible meaningful linear combinations (PCs) from a large number of variables [103]. Moreover, it permits the dimensionality reduction of data without a significant loss of useful information; exploration of hidden data structure; selecting the significant analytical signals; and clarifying the interrelation between samples and variables [102]. In terms of quantitative analysis of multivariate data, PCA is highly recommended as first instrument [104]. Compared to other multivariate regression tools which are not widely applied, PCA is the most beneficial tool [104,105].

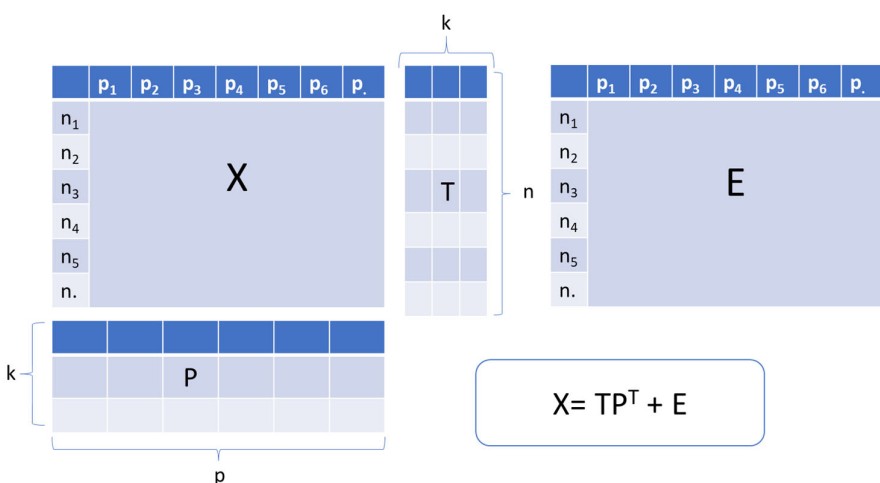

**Figure 2.** Schematic representation of Principal Component Analysis where X: raw data, T: Scores matrix, P: Loadings matrix, E: Residuals matrix. Adapted from Panchuk et al. [105] and Dupont et al. [106].

Partial Least Squares Regression

PLSR is an approach that generalizes and combines features from PCA and multiple regression to relate a descriptor matrix (called X) to a prediction vector/matrix (called Y). Afterwards, these two matrices are projected to a new space in order to find the common relations between them (X and Y), leading to a linear regression model. Thus, it aims to predict or analyze a set of dependent variables from a set of independent variables or predictors. In addition, PLSR is considered as a powerful linear regression method, capable of handling collinear variables, and accepting a huge number of variables [107,108]. PLS loadings are able to indicate a group of chemicals that co-vary with the given sample properties; also, by attributing qualitive results, PLSR can be used for discrimination [109,110].

Hierarchical Cluster Analysis

HCA is a method that involves the assessment of similarities between the samples based on their measured properties (variables). According to their adjacency in multidimensional space, the samples are grouped in clusters, and the results are shown in the form of dendrograms to facilitate the visualization of the relationships between the samples [111]. This method is suitable for posteriori data explorations [112]. Unlike PCA, HCA is frequently applied to determine similarities within several groups. These similarities are calculated using different possibilities, such as the correlation coefficient, the Euclidean distance, or the Mahalanobis distance [104].

3.3.2. Supervised Methods

Supervised techniques are based on the prior known data structures that are used to mold patterns and rules in order to predict new data. The advantage of supervised methods is the predictive capability of the models that can be easily used over a new sample. Supervised techniques can be performed by linear methods such as Partial Least Square Discriminant Analysis (PLS-DA), Linear Discriminant Analysis (LDA), or non-linear methods for instance Random Forest RF and Support Vector Machine (SVM), etc. [99,113].

Linear Methods

LDA uses the original variables as a basis in order to come up with a linear function, which maximizes the ratio of between-class variance and minimizes the ratio of within-class variance [114]. Since parameter optimization is not required, LDA is considered as a fast and powerful method to use in discriminant analysis [115].

PLS-DA is a combination of PLS regression and LDA, and is considered as the most popular supervised method used for classification in chemometrics [116]. By checking the behavior of variables, PLS-DA is able to afford excellent insights into the origin of discrimination. Furthermore, it can handle collinear data and is widely applied in modeling and biomarker discovery [117].

Non-Linear Methods

Kernel based models have been used to transform non-linear problems in the original data into a higher-dimensional feature space using particular functions called kernels. Subsequently, the non-linear problem becomes linear and can be solved readily [118]. kernel Fisher discriminant analysis (K-FDA) [119], kernel PLS (K-PLS) [120], and kernel OPLS (KO-PLS) [121] are kernel-based classification methods that have a high potential in solving non-linear problems.

SVM is a kernel-based classifier used to define decision boundaries and separate binary classes using support vectors [122]. This approach is centered on finding a hyperplane that splits two classes perfectly. Whereas the thickness of the margins is maximized, the distance of the plane to the data point is the closest for each class [123, 124]. SVM is more suitable for data of small sample sizes. However, SVM-based models suffer from the lack of transparency, and variable importance is difficult to obtain. In addition, this method does not provide a universal means of solving non-linear problems [125].

RF is an extremely efficient classifier for high-dimensional data. It is an ensemble-learning method that consists of a large number of classification and regression trees (CART) [126]. Classification trees are constructed based on training samples selected from the original samples by using a random resampling method with a replacement called bootstrapping. This latter is executed several times to build a large group of simple CARTs [127,128]. Compared to other classifiers, such as PLS-DA, RF has shown better performance, and could be an alternative to PLS-DA [129].

*3.4. Artificial Neural Networks (ANNs)*

A neuron is a cell that receives, manages, treats and delivers information using biochemical reactions. The human brain is a network formed by more than 10 billion interconnected neurons [130]. ANNs have been initiated as mathematical models simulating the nervous system. Simplified neurons were first introduced by McCulloh and Pitts, thus giving birth to neural networks [131].

Artificial Neural Networks can be defined as biologically inspired computer programs aiming to mimic information processes in the human brain. They are considered as a potential modeling technique, especially for data sets having a complex nonlinear relationship between dependent and independent variables [132]. ANNs are trained through knowledge collected from experiences used as inputs, unlike other techniques which perform specific tasks by simple implementation on a computer [133,134]. After training, new patterns can be used for a specific goal such as prediction and classification without a programming step [5]. Despite the fact that one neuron can execute simple information processes, the main advantage of ANNs, making them a powerful computational tool, is in connecting neurons in a network [135]. Because of the billion interconnections between neurons, ANNs can be easily used to recognize a large variety of input patterns. According to Ferentinos et al., ANN systems allow high-quality, rapid and high capacity of detection [136]. Moreover, they have been applied as a potential method for monitoring and assessment of environmental pollution [136,137].

### 3.4.1. ANN Architecture

ANNs are basically constructed of three layers; input, hidden and output layer. The first is made of a set of neurons used to receive external data and represent it to the neural network. In general, these inputs are normalized referring to the thresholds produced by activation functions. The hidden layer, also called the intermediate or invisible layer, is responsible for extracting patterns by analyzing processes and systems. The last layer of neurons is responsible for producing and presenting the final output [138–140]. Each layer is formed by artificial neurons called processing elements (PEs) that are interconnected, with links to simulate synapses. Each link or weight represents an adjustable coefficient used to moderate combination of input signals, transfer function and outputs [134,141]. Inputs are firstly multiplied by the weights, and then combined and passed through a transfer function to generate the output of that neuron. As biological neurons, artificial neurons can be excited or inhibited by the inputs. Excitatory inputs cause the summing mechanism of the next neuron to add, while the inhibitory inputs cause it to subtract [135]. Sigmoid function is the most frequent transfer function used, while a learning algorithm is used to adjust the weights in order to optimize the learning accuracy of the PE [142].

ANN can be composed of a single layer, leading to a simple neural network, or multiple layers leading to a multilayer neural network. The simple neural network is especially adapted for simple problems, while the multilayer neural network is principally used for more complicated processes [142]. A simple artificial neuron and a multilayered artificial neuron network schemes are illustrated in Figure 3 and Figure 4, respectively.

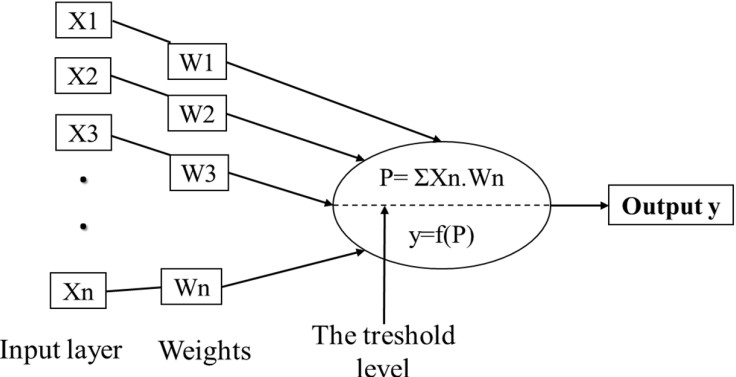

**Figure 3.** Architecture of a simple artificial neuron. Adapted from Agatonovic-Kustrin and Beresford [137] and Abraham [142].

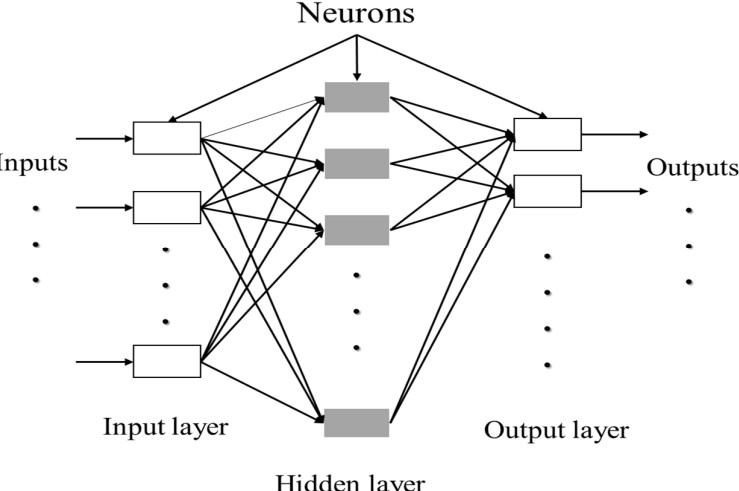

**Figure 4.** Architecture of a multilayered structure of artificial neural network. Adapted from Abraham [142].

ANNs' process and behavior are highly affected by neuron connection. Based on this, ANNs can also be categorized into two groups: feed-forward networks and feedback or recurrent networks [143].

Feed-forward networks are considered as a static system; there are no loops (absence of feedback) or connections from output to input neurons. The signals are transmitted in one direction only from inputs to outputs. Therefore,outputs of the previous operation are not memorized and the next one depends only on the input signals (memory-less) (Figure5) [144,145]. This type of architecture can be structured as monolayer or multilayer composed of one or more hidden neural layers [137,140].

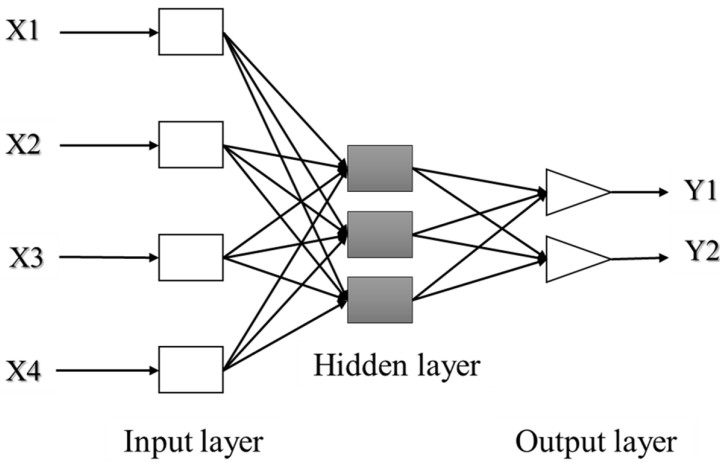

**Figure 5.** Feed-forward network structure. Adapted from Agatonovic-Kustrin and Beresford [137].

Recurrent (or feedback) networks are considered as a dynamic system where loops are formed as a result of connections from output to input neurons(loops). In this case, the outputs of neurons are used as feedback inputs for other neurons. Such a network system memorizes the previous state, so that the next state depends not only on input signals but also on the previous outputs (Figure 6) [135].

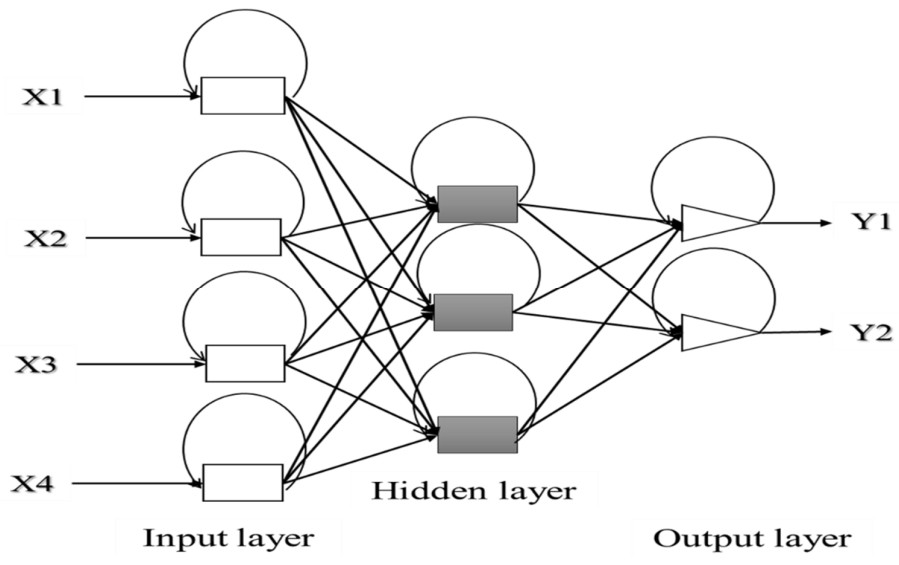

**Figure 6.** Feedback network structure. Adapted from Agatonovic-Kustrin and Beresford [137].

Other architectures have also been also described in the literature, such as the Elman network, adaptive resonance theory maps, and competitive networks. In general, the architecture is selected based on the properties and requirements of the application [137]. In parallel, following the type of network and training algorithm, different activation functions may be used: logistic sigmoid, linear, threshold, Gaussian or hyperbolic tangent functions [146].

### 3.4.2. Learning Rules

After selecting the adequate network, a training step is required to produce the desired outputs at the suitable time, hence the importance of adopting a restrictive learning rule. Learning rules are different from ANN models; ANN models represent the

arrangement and the disposition of neural networks, while algorithms compute the output. A learning rule is defined as any systematic adjustment of weights that minimizes error function or maximizes a specified benefit function. This means the ANN's recognizing abilities are based mainly on weighted links [147].

Two main rules can be adopted for training: supervised and unsupervised algorithms [135]. Fully connected, supervised ANNs with back propagation learning rule are the most popular, in particular for prediction and classification. In parallel, the Kohonen or Self Organizing Map with unsupervised learning algorithm is highly used for finding complex relationships among data [137].

**Supervised algorithms:** This rule is based on the relation between the inputs and outputs of a training set. By using this type of algorithm, the network generates the desired output for every input data. Errors between the obtained values and the known ones in each output layer nodes are first calculated. Then, these errors are used by the learning rule to determine proper weights giving suitable results and a high quality of prediction [148]. It has been reported that the number of hidden neurons must be optimized, otherwise a system failure can occur decreasing prediction abilities [137,148].

**Unsupervised algorithms:** This rule requires a training task to classify input patterns into different categories according to the correlation between them [149]. It consists of an auto-organized system, since features are selected by the system itself to categorize the input data and achieve a specific classification. This system behavior implies competition between neurons, co-operation, or both [137,142,148]. For competitive learning, among a group of neurons, the one responding strongly to the input will inhibit the other neurons. For co-operative learning, neurons of each category work together to enhance their outputs [137].

**Hybrid learning** is a mixture of supervised and unsupervised learning, which are combined to determine weights [140].

**Reinforcement learning**: This type is a variant of supervised learning based on the correctness of network outputs. This learning rule requires a reward for a correct output and a penalty for the wrong one [140,150]. Trial and error research and delayed reward are the two major special features of reinforcement learning [142].

**Error-correction rules:** During the learning process, an error correction rule is used to calculate the difference between the desired and the obtainedvalues. The aim is to adjust the connection weights and gradually reduce this error [149].

**Hebbian rule:** Almost all neural networks' learning techniques are considered as a variant of the Hebbian learning rule, since it is the oldest learning rule based on the observations of neurobiological experiments [151]. The interconnection of two neurons has to be enhanced if the two neurons are stimulated at the same time. However, in the case of a single layer, one of the interconnected neurons will constitute the input unit while a second one forms the output unit [142].

3.4.3. Training and Testing Neural Networks:

The best way to ensure the efficiency of ANNs is to execute a number of problems with different characteristics. The adjustment process or training aim to adjust weights and threshold to obtain output values close to the desired ones. The number of these trainings is determined by the complexity of the samples. During training, some noises can be added to enhance the system applicability in real samples [148]. Poor training data generates an uncertain network, while an overtraining generates a system unable to make execution outside of the training set. Other parameters have to be considered, such as the number of nodes in the hidden layer which plays a key role in the network functionality. Using many nodes in the hidden layer will increase the number of possible computations of the algorithm, while just a few nodes in the hidden layer can prevent the algorithm from learning. Therefore, the right balance needs to be picked [146]. In parallel, selecting initial weights is crucial, in particular for multidimensional ones. This can be obtained by testing different initial weight values to attain the desired results. Finally, the system

performance is also affected by the learning rate since it controls the size of each step [142,148].

*3.5. Modern Computational Approaches:*

These approaches are widely used to screen and classify the design/shape of bioreceptors and forecast the resultant interaction. Since computational approaches are based on computational calculations, they have anultra-sensitive and selective potential. In this context, computational docking and molecular dynamics (MD) are two common techniques based on numerous algorithms and used to predict the interrelation receptor–ligand.

Computational docking is a promising approach aiming to enrich experimental structural data on receptor–ligand interactions and optimize the conformation and the quaternary structure of the formed complexes [152,153]. These techniques are based on placing a small molecule in the specific binding site of its macromolecular target with an estimation of the binding affinities [154]. The docking process involves two main approaches; sampling and scoring, that can be coupled together or occur in different stages [155]. Sampling is a search process that generates different possibilities of binding orientations and/or conformations (i.e., modes) between two molecules within the constraints of the receptor binding site. Scoring is the computation, using the score between two molecules in a binding mode. Afterwards, the binding modes are ranked according to their binding scores, and the best ones can be selected as the final docking solutions (ligand conformation, orientation, and translation) [156,157].

Molecular dynamics is an established simulation tool in biomolecular study, offering information about the hydrodynamic behavior [158,159]. It is based on modeling molecular structure using potential-energy functions. In other words, MD simulations involve the iterative numerical calculation of instantaneous forces present in the system, a set of particles that move in response to their interactions according to a defined equation, and the consequential movements in that system. This approach is principally used to obtain data in the precision and evolution of molecular conformations, but also kinetic and thermodynamics information [160]. Moreover, this dynamic simulation method is used to clarify the interaction mechanism quantitatively and qualitatively [161]. In term of environmental monitoring, these computational techniques have been used to redesign and model essential enzymes (organo-phosphatase) for pesticide hydrolysis, to reactivate their unexploited catalytic potential, and to determine their mode of action [162,163].

## 4. Application of Artificial Intelligence in Environmental Biosensing

Monitoring the environment via biosensors is based on two approaches; affinity-based sensing where pollutants can be detected by a highly specific receptors, i.e., antibodies or aptamers which may come at a cost; and inhibition sensors, particularly based on enzymes that are inhibited by pollutants [164]. The enzymatic approach is much simpler and less expensive but the receptor may be affected by other pollutants, influencing sensitivity and selectivity. Therefore, sensors based on more than one bio-receptor have to be used for the simultaneous detection of different analytes. To achieve this, the cross-disciplinary concept of AI is more than a necessity. AI-based biosensors are based on three elements: information collection, signal conversion and AI-data processing. Information collection refers to regularly monitoring physical, chemical, biological, or environmental information using biosensors. The signal conversion aims to generate an electric output, with a defined sensitivity, from the collected information. Finally, AI-data processing comprises interface, data classification, data model and analysis, and decision layer [165,166].

### 4.1. Environmental Monitoring Based on Chemometric Methods

Chemometric methods are extensively used in environmental monitoring as they permit the identification and description of the interrelations between environmental factors. In addition, they clarify the potential impact of these factors on the environment [167].

A chemometric approach was used to get a better insight into some trace metal patterns. The concentrations of Cu, Zn, Mn, Fe, K, Ca, Mg, Al, Ba and B in 26 herbal drugs were studied using flame atomic absorption and emission spectrometry, as well as inductively coupled plasma atomic emission spectrometry. Afterwards, a PCA was used to highlight the relation between the elements. Four significant factors were identified and partly attributed to the significant influential sources and high mobility of some elements, thus referring also to potential anthropogenic contamination. In this work, of all chemometric methods the PCA is the most suitable because it allowed the identification of factors that are substantially meaningful [168]. In contrast to the previous work which was based on one chemometric technique, this research group used three chemometric techniques (PCA, LDA and cluster analysis). In this study, they evaluated trace metal concentrations (As, Ba, Ca, Cd, Co, Cr, Cu, Fe, Mg, Mn, Ni, Sr and Zn) in spices (black pepper, chili pepper, cinnamon, cumin, sweet red pepper and turmeric) and herbs (mint, thyme and rosemary). They used an atomic spectroscopy for the determination of trace metals, and chemometric evaluation for the classification study. As a result, herbs and spices were classified into five groups by PCA and CA. Compared to the results obtained by LDA, it was found that all group members determined by PCA and CA are in the predicted group and that 100% of original grouped cases are correctly classified by LDA [169]. Chemometric methods are also used to evaluate phenolic compounds. In this context, phenolic and flavonoid compounds are identified in four unifloral honey types using chemometric approaches (PCA and HCA). Moreover, the correlation among the physico–chemical characteristics was also studied. The first three PCs explained more than 83% of the variance with minerals showing the highest discriminating power while HCA successfully classified all the unifloral honey samples. This work demonstrated that it is possible to effectively classify honey by applying chemometric techniques (PCA and HCA) [170].

Nowadays, near-infrared spectroscopy (NIRS) is combined with chemometric techniques in order to monitor the environment [171]. A study reported the quantification of pharmaceuticals in wastewater using Fourier transformnear-infrared (FT-NIR) spectroscopy methodology. The samples were treated by chemometric techniques in order to develop and validate the quantification models. The obtained results were found adequate for the prediction ofibuprofen, sulfamethoxazole, 17β-estradiol andcarbamazepinewith $R^2$ around 0.95 and residual prediction deviation values above four. FT-NIR spectroscopy is not used as a direct analysis technique because it suffers from the complexity of the spectra. For this reason the FT-NIR methodology is combined with chemometrics, exploring a promising technology which is able to quantify a wide range of organic compounds [172]. Since lichens are extremely sensitive to the presence of substances that alter atmospheric composition, near-infrared (NIR) spectroscopy was developed as a tool for discriminating between lichen samples exposed to air pollution. In addition, PCA and LDA are used as chemometric methods to successfully discriminate between samples from polluted and non-polluted areas. The PCA was applied in the NIR spectra as a multivariate display method to visualize the NIR data. The LDA was carried out to discriminate between lichens based on their exposure to pollutants. On average, 95.2% of samples were correctly classified, and 100.0% external prediction was achieved. These results showed that NIR-spectroscopy based chemometric methods can be considered as a potential reliable method to support traditional methods for the discrimination between lichen samples according to their exposure to pollutants[173]. Another study based on NIR spectroscopy combined with chemometric techniques was also described as a monitoring tool of exhaust air from poultry operation systems.

Samples were collected from the exhaust air of two poultry houses using sophisticated filter sampling protocols. This study aimed to monitor spectral differences caused by the cleaning device, and to follow changes in exhaust air characteristics during a fattening period. PCA, LDA, and FA were successfully used to classify the NIR exhaust air spectra according to fattening day and origin. The results show that the dust load and the composition of exhaust air significantly affect the NIR spectral patterns. These results confirmed that the sample classification according to fattening days, raw and cleaned exhaust air was possible by using chemometric methods [174]. These results demonstrate the high potential of NIRS combined with chemometric techniques to monitor an environment in a non-destructive, quick, and elegant way.

Chemometric approaches are also used to evaluate the influence of seasonal changes in volatile organic compound concentrations [175], to rapidly classify heavy metal-exposed freshwater bacteria [176], and to monitor air and water quality [177–180].

The combination of chemometrics and sensors is positioned to be a major breakthrough in collecting meaningful data and forits depth of analysis in determining trends and interrelationships [167,171,181,182]. In this context, Lamagna et al., used an electronic nose to analyze samples that were collected above a river basin in Argentina. This electronic nose consisted of 32 polymer sensors used to measure the concentration of $SO_2$ and $H_2S$ in the air. The combination of PCA and the sensor was used to identify the sampling sites that deviated significantly from a control sample of clean air. This combination was able to identify a correlation that existed between polluted sampling sites and the electronic nose response, indicating that this system could be used for monitoring pollution in rivers [183]. Another study reported the use of a potentiometricmultisensor systemin order to assess water toxicity in terms of the bioassay with three living test organisms:*Daphniamagna*,*Chlorella vulgaris*and*Paramecium caudatum.*Using a PLS regression from the obtained data, the prediction of water toxicity with relative errors 15–26% was attainable. Further experimental work with larger data sets is required to improve the performance of such "artificial bioassays" [184].

### 4.2. Environmental Monitoring Based on Artificial Neural Networks (ANNs)

Developing a biosensing system able to detect and quantify several targets simultaneously represents a challenging tool in environmental monitoring. It has been reported that various pesticides, including organophosphorus (OPs) and carbamate insecticides, can inhibit cholinesterase activity. Based on this, several enzymatic inhibition biosensors have been developed [185–192]. However, all have the difficulty in discriminating between different inhibitors. To overcome this problem, biosensing platforms have been conjugated with an ANN to find a pattern relating inhibitor concentrations to the observed inhibition percentages, allowing thus an accurate biosensing [193]. Table 1 summarize some application of ANN in environmental monitoring.

**Table 1.** Summary of Artificial Neural Network (ANN)-based biosensing platforms applied to pollutant detection.

| Biomarker | DetectionMethod | Limit of Detection (LOD) | Prediction Error | Correlation Coefficient | Reference |
|---|---|---|---|---|---|
| Chlorpyrifos-oxon and chlorfenvinfos | Amperometric | Chlorpyriphos-oxon 1 g/L Chlorfenvinphos 0.004 g/L | ANN1 4.4% ANN2 0.23% | >0.986 | [194] |
| Paraoxon and Carbofuran | Amperometric | $2 \times 10^{-7}$ g/L | paraoxon 0.9 mg/L | / | [195] |

| | | | | carbofuran 1.4 mg/L | |
|---|---|---|---|---|---|
| Chlorpyriphos-oxon (CPO), Chlorfenvinphos (CFV), and Azinphos-methyloxon (AZMO) | Amperometric | CPO 1.55 × $10^{-6}$ g/L CFV 17 × $10^{-6}$ g/L AZMO 2.44 × $10^{-6}$ g/L | CPO 1.82% CFV 1.51% AZMO 2.3% | CPO 0.985 CFV 0.991 AZMO 0.997 | [196] |
| paraoxon with malaoxon or carbofuran | Amperometric | 0.5 × $10^{-6}$ g/L | paraoxon 0.4 µg/L and carbofuran 0.5 µg/L Malaoxon 0.9 µg/L and paraoxon 1.6 µg/L | / | [197] |
| chlorpyriphos, dichlorvos and carbofuran | Spectrophotometric | / | / | chlorpyriphos 0.916 Dichlorvos 0.991 Carbofuran 0.959 | [198] |
| Dichlorvos and Carbofuran | Amperometric | Dichlorvos 1.7 × $10^{-7}$ g/L carbofuran 9.07 × $10^{-7}$ g/L | dichlorvos 0.3 nM carbofuran 4.0 nM | >0.918 | [199] |
| paraoxon, dichorlvos, and carbofuran | Amperometric | Paraoxon 1.07 × $10^{-5}$ g/L Dichorlvos 1.39 × $10^{-8}$ g/L Carbofuran 1.29 × $10^{-7}$ g/L | Paraoxon 3.38% Dichorlvos 6.25% Carbofuran 11.06% | >0.970 | [200] |
| Dichlorvos and Methylparaoxon | Amperometric + Flow Injection Analysis (FIA) | / | Dichlorvos 0.0093µM methylparaoxon 0.18 µM | Dichlorvos 0.879 methylparaoxon 0.889 | [201] |
| parathion-methyl (PTM), fenitrothion (FT) and parathion (PT) | Differential Pulse Stripping Voltammetry (DPSV) | PTM 4.8 × $10^{-6}$ g/L FT et PT 4.5 × $10^{-6}$ g/L | <10% | 0.999 | [202] |
| 4-aminophenol, 4-chlorophenol, and 4-chloro-3-methylphenol | Electrochemical | / | / | 4-aminophenol 0.972 4-chlorophenol 0.998 4-chlorom-3-methylphenol 0.968 | [203] |

| | | | | | |
|---|---|---|---|---|---|
| Atrazine and Simazine | Total InternalReflectance Fluorescence (TIRF) | Atrazine 0.2 × $10^{-3}$ g/L Simazine 0.3 × $10^{-3}$ g/L | / | / | [204] |
| Cd, Cr, Cu, Mo, Ni, Pb, Sb, Tl, and Zn | Modeling | / | 0.04 μg kg$^{-1}$ (Cd) to 0.1 μg kg$^{-1}$ (Cr) | / | [205] |
| As, Cd, Fe, Hg, Pb, S, and Sb | Reflectance spectroscopy | / | <0.0965 | >0.845 | [206] |
| Pb$^{2+}$, Cd$^{2+}$ and Cu$^{2+}$ | Potentiometric | $10^{-3}$ g/L | Pb$^{2+}$ 3.0 % Cd$^{2+}$ 4.1 % Cu$^{2+}$ 5.2 % | >0.975 | [207] |
| Pb(II), Cr(VI), Cu(II), Cd(II), Cl$^-$ | potentiometric + flow injection system (FIS) | Pb (II) 6.2 × $10^{-4}$ g/L Cl$^-$ (alone) 3.5 × $10^{-5}$ g/L. | all metals 10–15% Cl$^-$ 30% | / | [208] |
| Cd(II), Pb(II) and Hg(II) | Differential Pulse Anodic Stripping Voltammetry (DPASV) | / | Pb(II) 14.1 ppb Hg(II) 21.71 ppb Cd(II) 23.17 ppb | Pb(II) 0.998 Hg(II) 0.999 Cd(II) 0.964 | [209] |
| Pb$^{2+}$, Cd$^{2+}$, Cu$^{2+}$ and Zn$^{2+}$ | Potentiometric | / | / | >0.948 | [210] |
| Cd$^{2+}$, Cu$^{2+}$, Pb$^{2+}$ | Potentiometric + FIA | Cd$^{2+}$ 6.18 × $10^{-10}$ g/L Cu$^{2+}$ 3.937 × $10^{-10}$ g/L Pb$^{2+}$ 1.16 × $10^{-9}$ g/L | Cd$^{2+}$ 2.7 μM Cu$^{2+}$ 0.1 μM Pb$^{2+}$ 0.55 μM | Cd$^{2+}$ 0.91 Cu$^{2+}$ 0.972 Pb$^{2+}$ 0.991 | [211] |
| Pb$^{2+}$, Cd$^{2+}$, Zn$^{2+}$ | Potentiometric | / | 15–30% | / | [212] |
| binary (Cu$^{2+}$/Pb$^{2+}$), (Cu$^{2+}$/Zn$^{2+}$) ternary (Cu$^{2+}$/Pb$^{2+}$/Zn$^{2+}$), (Cu$^{2+}$/Zn$^{2+}$/Cd$^{2+}$) | Potentiometric + FIA | / | <7% | >0.954 | [213] |
| Zn$^{2+}$, Cu$^{2+}$, Ni$^{2+}$ | Optical | $10^{-7}$ mol L$^{-1}$ | Zn (II) 0.06 ± 0.10 M Cu (II) 0.05 ± 0.07 M Ni (II) 0.02 ± 0.03 M | 0.99 | [214] |
| Ammonium NH$_4^+$; potassium K$^+$ | Potentiometric | / | App 1% | NH$_4^+$ 0.999 K$^+$ 0.998 | [215] |

| | | | | | |
|---|---|---|---|---|---|
| Phenol, cathecol and m-cresol | Ampérometric biosensors | / | <0.010 Mm | Phenol 0.988; Cathecol 0.997; m-cresol 0.993 | [165] |
| catechol (CC) and hydroquinone (HQ) | simultaneouskineticdetermination+FIA | HQ $5 \times 10^{-5}$ g/L CC. $7 \times 10^{-5}$ g/L | <5% | HQ 0.994 CC 0.989 | [216] |
| Caffeicacid and Catechol | Amperometric | / | <0.5% | >0.999 | [217] |
| Catechol (phenolic compound) | Electrochemical | $3.52 \times 10^{-6}$ g/L | <0.4% | 0.99 | [218] |
| Catechol,*m*-cresol and guaiacol | Voltametric | $8 \times 10^{-2}$ g/L | / | Catechol 0.952; m-cresol0.977; guaiacol 0.886 | [219] |
| 2,4-dinitrophenol, 4-nitrophenol, and picric | Voltametric | / | 0.076 | >0.948 | [220] |
| imidacloprid, albendazole, fenbendazole, praziquantel, fipronil and permethrin | liquidchromatography | imidaclopril, albendazole and fenbendazole $4 \times 10^{-6}$ g/L praziquantel and fipronil 8 $\times 10^{-6}$ g/L permethrin 2.6 $\times 10^{-5}$ g/L. | / | / | [221] |
| phenol, 2 chlorophenol, 3-chlorophenol and 4-chlorophenol | kineticspectrophotometric | / | <5% | / | [222] |
| Myclobutanil | homogeneous liquid–liquid microextraction + gas chromatography-mass spectrometry | / | 0.89% | 0.990 | [223] |
| Ochratoxin A (OTA) | Spectrofluorimetric | / | 0.036 | 0.995 | [79] |
| Aflatoxin B1 (AFB1) | Potentiometric | / | 0.0026% | >0.999 | [224] |
| microcystin (MC)variants mixture: MC- | Enzyme-kinetic | MC-LR 21.2pM | MC-LR 2.76–10.0% | MC-LR 0.996 MC-YR 0.983 | [225] |

| LR, and MC- | MC-YR 0.26– |
| YR | 19.78% |

### 4.2.1. Pesticide Monitoring

Several intelligent biosensors based on the principle of acetylcholinesterase (AChE) inhibition and chemometric data analysis using ANNs have been developed [136,194,196,197,217,226]. Istamboulie et al. developed an amperometric acetylcholinesterase biosensor based on ANNs to selectively quantify and discriminate a mixture of pesticides in real water samples using two constructed ANNs. They modeled the combined response of two pesticides (chlorpyrifos oxon and chlorfenvinfos) using sensors incorporating wild-type electric eel AChE and drosophila mutant AChE, associated or not with a phosphotriesterase PTE. A satisfying prediction ability was obtained with correlation coefficients better than 0.986 and a limit of detection of 0.02nM for chlorpyriphos-oxon, and 0.15 nM for chlorfenvinphos [226].

This developed tool used two enzyme variants to detect only two pesticides. Therefore, other biosensors have been described to reduce the number of enzyme variants and detect more analytes. A further investigation was conducted by the same group to generate an array of three biosensors formed by screen printed carbon electrodes and two different AChE enzymes. The developed device was able to discriminate various insecticides with high accuracy. In this work, ANNs were used in combination with the enzymatic activity rate as analytical signal, rather than the inhibition percentage.The simultaneous detection of the three insecticides mixtures (chlorpyriphos-oxon, chlorfenvinphos, and azinphos-methyl-oxon) was achieved successfully by applying only two AChEs from *Drosophila melanogaster* (wild-type and genetically modified). The obtained results were very interesting in terms of sensitivity and precision with low error, and correlation coefficient higher than 0.985. As compared to the previously discussed work, this method is faster and simpler, allowing the detection of three compounds [196]. Based on the same principle and using a spectrophotometric assay with three different enzyme systems, the detection of more than two pesticides has been carried out. This system was successfully applied to resolve the three insecticides chlorpyriphos, dichlorvos and carbofuran in real water samples, with high correlation coefficients of, respectively, 0.916; 0.991; 0.959 [198].

Later, Bachmann and Schmid developed, for the first time, a highly sensitive screen-printed amperometric multielectrode biosensor for the rapid quantification and discrimination of paraoxon and carbofuran in insecticides mixture. They used four types of native or recombinant AChEs as specific ligands by combining the AChE-multisensor with feed-forward ANNs for quantitative inhibition analysis of the binary mixture. The prediction errors were 0.9 mg/l for paraoxon and 1.4 mg/l for carbofuran with apparent Michaelis–Menten constant, *Km* of 0.08 mM. This multi-sensor achieved a LOD of 0.2 μg/L which is much lower than that obtained in other reports [195].

In another report, Schäfer et al. described a high potential assay for the specific detection of organophosphorus compounds and carbamates incorporating extensive chemometric data analysis using soluble AChE in the microtiter plate method. In contrast to the biosensors described above, this assay procedure is restricted to laboratory use only [227].

Crew et al. went a step further and presented a portable analytical system integrating an array of six AChE-based enzymatic biosensors in a novel automated device equipped with an efficient ANN program. This analytical tool was used to successfully resolve a mixture of 6 pesticides, dichlorvos, malaoxon, chlorpyrifos-oxon, chlorpyrifos-methyl-oxon, chlorfenvinphos and pirimiphos-methyl-oxon from nM to mM concentration range, in less than 6 min. In addition, the system was also validated for different sample matrices, including water, food and vegetable extracts, without false positives or negatives (Figure 7) [228]. Finally, several reports have also described the ANN-assisted biosensing of

dipterex, dichlorvos, methyl paraoxon, and omethoate [194,196,201,229]. A cell-based biosensor incorporating ANNs was also established to classify pesticides into different groups (pyrethroids, carbamates and OPs) [136].

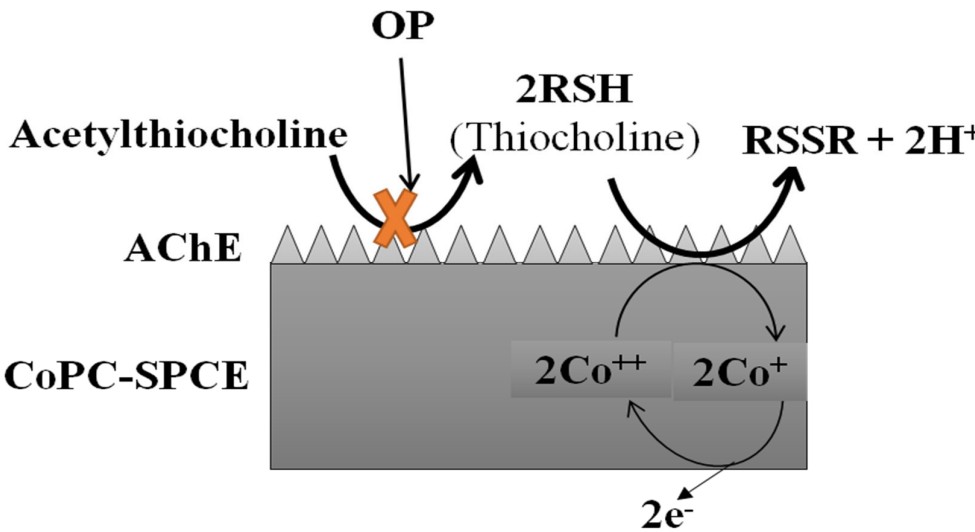

**Figure 7.** Schematic diagram explaining the principle of the reactions based on this amperometric biosensor. Adapted from Crew et al. [228].

### 4.2.2. Heavy Metals Monitoring

In addition to pesticides, heavy metals biosensing constitutes a great challenge in environmental monitoring. ANNs have been used for soil and water studies, such as predicting contamination, measuring organic content and macronutrients, pollutant infiltration properties and soil classification [205,206,214,230–234].

A sorption model was developed by Anagu al. for the estimation of heavy metals sorption from basic soil properties and evaluating the risks related to their apparition. In this work, nine heavy metals have been targeted; Cd, Cr, Cu, Mo, Ni, Pb, Sb, Tl, and Zn. The ANN models showed a high performance with a root mean square error (RMSE) ranging from 0.04 µg.kg$^{-1}$(Cd) to 0.1µg.kg$^{-1}$(Cr) and a modelling efficiency (EF) ranging from 0.79 (Cr) to 0.94 (Cd, Zn). By comparing these results to those based on multiple linear regression (MLR), we can conclude that ANN models behave better than MLR, where EF ranged from 0.03 to 0.13 [205].

ANNs have also been combined with electrochemical biosensing platforms for HM detection in the environment, by using various types of electrodes. For instance, an electronic tongue approach based on polyvinyl chloride (PVC) membranes has been used in the rapid and simple on-site monitoring of several heavy metals [210,235–237]. In this context, a mixture of three heavy metals (Pb$^{2+}$, Cd$^{2+}$and Cu$^{2+}$) have been resolved using an e-tongue (ANNs) integrated with a membrane selective electrode. A limit of detection of 1mg·L$^{-1}$was attained for the three targets with a good reproducibility. In addition, the correlation was significant for the three ions, exceeding 0.975. Finally, this system showed a good prediction ability when applied to contaminated soil samples [207]. Despite the interesting resulting, this strategy is adapted only for three HMs. Therefore, another group investigated the simultaneous detection of a mixture of four HMs by using the same principle of potentiometric e-tongue and PVC membranes using ANNs. The method was successfully applied for the quantification of low levels of Cu$^{2+}$, Pb$^{2+}$, Zn$^{2+}$, and Cd$^{2+}$ ions from quaternary mixtures in soil, open-air waste streams and rivers with low root mean squared error values (~1 mmol·L$^{-1}$) [210].

Besides PVC membranes, other electrodes have been used in ANN-based sensing of heavy metals. A research group investigated the resolution of a mixture of Pb, Cr, Cu, Cd and chloride ions by using a flow injection set-up method with multiple chalcogenide glass chemical sensors (electrodes). For this, seven electrodes were constructed and the output signals were analyzed using multivariate analysis method including artificial neural networks. The detection limit for Pb was about 3 μM and about 1 μM for the other metals. Besides, the chloride ions can be determined down to about 1 μM alone, and 20 μM simultaneously with other pollutants. The average error for metals resolution in the scale from the detection limit went up to about 3 mM, with 10–15% for the metal mixture, and about 30% for Cl⁻ [208]. In another report, a graphite epoxy composite electrode wasused to detect a quaternary HMs mixture using a specific type of voltametric detection method. In this work, for the first time, a three-sensor array was applied for the simultaneous quantification of Cd (II), Pb (II) and Hg (II) ions in certified samples. They used two graphite epoxy composite electrodes modified with carboxybenzo (CB)-18-crown-6 and CB-15-crown-5, respectively, and an unmodified one. The crown ethers serve as molecular collector with ability to selectively coordinate with the metal ions for complex formation by means of ion-dipole interaction with metal ions. The expected and the obtained results were very close and the achieved LODs and limits of quantification (LOQs) were at levels of μg·L⁻¹. These results confirmed that this biosensing array provides a discrimination power to resolve HM mixtures [209]. Other studies devoted to the application of crown ether-modified electrodes for the simultaneous determination of individual/mixture HMs have also been reported [238–242].

Finally, electrochemical biosensing technologies based on ANNs seem to be successfully adapted for the simultaneous detection of EP mixtures in the environment at the ultra-trace level. Furthermore, these systems provide several advantages such as high specificity, fast response time, portability, simplicity, and low cost,making them reliable devices for EP monitoring [243,244].

4.2.3. Air and Water Quality Monitoring

Several approaches based on ANNs have been developed for modeling the monitoring of air and water quality or of environmental systems in general [245–251]. In this context, a statistical model was developed for the accurate forecasting of the Air Pollution Index (API) in industrial and residential monitoring stations in Malaysia. For this, the autoregressive integrated moving average (ARIMA), fuzzy time series (FTS) and artificial neural network were employed. By comparing the obtained results with that of ARIMA and FTS, the ANNs exhibited the smallest error in forecasting API values. This method thus constitutesan effective way to control the air quality and for decision-making processes [252]. Later, Wang et al. [253] studied the causal relationship between urbanization and water quality indices, and used this as a support to predict the water quality. Correlation and path analysis were used to identify the causal relationships, followed by a back-propagation neural network to predict water quality. The obtained coefficients of correlation were higher than 0.76 improving thus the performance of the optimized model for predicting urban water quality with nonlinear variation. However, this work was limited to simulation studies in calculating the API. Another report describing experimental validation of modeling has been published. The authors developed an electrochemical sensor array based on the inhibition of immobilized bacteria for the preliminary detection of a wide range of organic and inorganic pollutants. The objective of this work was based on heavy metal salts, pesticides, and petrochemicals monitoring in water using three types of bacteria, namely *Escherichia coli*, *Shewanellaoneidensis*, and *Methylococcuscapsulatus*. The response of each pollutant was monitored with more accurate recognition using the ANNs. The obtained system was capable of resolving all these pollutants at low concentrations (down to 0.1 μM) [254].

### 4.2.4. Pharmaceuticals Monitoring

Several electrochemical biosensors assisted by ANNs have been developed for therapeutic substances detection, in particular phenolic compounds mainly based on tyrosinase (Tyr) and laccase (Lac) [255]. Gutéet al. developed an electrochemical biosensor combined with ANNs to simultaneously detect different phenols (phenol, catechol and m-cresol). A single graphite epoxy bio composite bulk-modified with Tyr was used in combination with a sequential injection analysis (SIA) system. Departure information was extracted via the whole voltammogram, and then ANNs were used for extraction and quantification of each compound. The model of the three analytes shows a good correlation coefficient: phenol 0.988; catechol 0.997; and m-cresol 0.993 [165]. In this study, the authors compared the obtained results with that of a previous study reported by Trojanowicz et al. [256], based on the same principle, by using only the steady-state responses rather than the whole voltammogram. The authors confirm that the results of this method were not performant with low correlation coefficients between expected and predicted values. Therefore, they attributed this to the lack of the data afforded by the system. Later, another biosensor based on the same principle and on three composite electrodes bulk-modified with tyr, lac, and copper nanoparticles has been developed for the simultaneous monitoring of catechol,*m*-cresol and guaiacol in wastewater. The method was based on a voltametric bioelectronic tongue where the electrochemical responses of the three composite electrodes were compressed by means of fast Fourier transformation. Each voltammogram was compressed down to a number of coefficients which were used as inputs to the ANN model. The system was also applied for the photo-degradation monitoring of the three phenolic pollutants. This proposed approach exhibited a good correlation coefficient with a RMSE of 1.50 for the training subset and 4.20 for the testing subset. These results showed the powerful potential of this approach for the speciation of different phenolic compounds in wastewater and the monitoring of its mineralization. Moreover, using this approach, a quantitative multi-determination of a mixture of chemical species can be easily attainable with simple equipment, shifting the complexity from the sensors to the software side [219]. Later, Boroumand et al. reported the flow injection simultaneous kinetic determination of two isomers, Hydroquinone and catechol. In this work, a simple FIA manifold equipped with double injectors and single detection system was combined with a Bayesian Regularized Artificial Neural Network for determining the targets in real samples. Detection limits of 0.05 and 0.07 mg·L$^{-1}$ were obtained for HQ and CC, respectively [216].

### 4.2.5. Biotoxins Monitoring

The great potential of ANNs for resolving multivariate calibration problems has been also explored for biotoxins analysis. For instance, Saidi et al. reported the rapid determination of OTA by a spectrofluorimetric procedure-based ANN. The study described the application of the ANN method to a set of spectrofluorimetric data obtained from the analysis of wheat and rice samples contaminated with OTA. The obtained results were significant with a correlation coefficient of 0.995 and a mean square error of 0.036 [79]. Another method using an electronic nose equipped with a 10-metal oxide sensor array has been subsequently reported based on a set of data collected over five years. This e-nose array allowed the rapid identification of aflatoxin B1 and fumonisins in maize samples using three different statistical approaches: ANN, logistic regression, and discriminate analysis. Notably, ANN gave better results than the other methods, with 78% and 77% accuracy for AFB1 and FBs, respectively. In contrast to certain strategies [257], the majority of the data set in this work were collected over five years and analyzed in different ways providing, thus, a remarkable reliability [258].

Another model based on artificial neural network was developed to estimate blue-green algae fluorescence for the year-round data collected in 2016–2017 from western Lake Erie, USA. Eight input parameters including phosphorous, nitrogen, chlorophyll-a, air

temperature, water temperature, turbidity, wind speed, and pH were used to run the model. Five different learning algorithms were tested, and among these the Levenberg-Marquardt algorithm gave the highest $R^2$ values of 0.98 and 0.72 for the eight and three (phosphorous, nitrogen, and chlorophyll-a) input parameters, respectively. Based on this, the best estimation of blue-green algae fluorescence was achieved using eight input parameters while a weak correlation was obtained with the three input parameters. This method allows the rapid and low cost prediction of blue-green algae by using simple measurements [259].

### 4.3. Environmental Monitoring Based on Genetic Algorithm (GA)

Genetic algorithm plays an important role in environmental monitoring, particularly, in optimizing the quality of the developed systems, improving thus the final results [260].Table 2 summarize some application of GA in environmental monitoring.

**Table 2.** Summary of genetic algorithm (GA) applied to pollutants detection.

| Biomarker | Detectio Method | LOD | Predictionerr or | Correlation Coefficient | Reference |
|---|---|---|---|---|---|
| Malathion | Fourier transform infrared + partial least square (PLS) | / | 0.059 mg·mL⁻¹ | 0.999 | [261] |
| chlorpyrifos contents | Surface-enhanced Raman spectroscopy (SERS) + PLS | / | 0.29 | 0.96 | [262] |
| Zinc + Copper | Spectrophoto metry + PLS | / | Copper 0.0407 Zinc 0.0865 | 0.999 | [263] |
| COD, BOD5, TSS, P, TN, NO3 −N | spectrophoto metric | / | <4% | / | [264] |
| Ochratoxin-A (OTA) and aflatoxin-B1 (AFT-B1) | Surface-enhanced Raman spectroscopy (SERS) + PLS | OTA 2.63 × 10⁻⁹ g/L AFT-B1 4.15 × 10⁻⁹ g/L | OTA: 0.456 µg·mL⁻¹ AFT-B1: 0.441 µg·mL⁻¹ | OTA: 0.962 AFT B1: 0.972 | [265] |

Karkraet al. have applied genetic algorithm to 24 water samples containing eight different heavy metal ions (Cd, Co, Zn, Ni, Cu, Cr, Ag and Pb). The goal of this study consisted in selecting the optimal electrode as well as the corresponding frequency allowing the classification of heavy metal ions. Different electrodes including gold, platinum, glassy carbon and silver nanoparticles were tested in combination with different frequencies. Genetic algorithm was used to select the electrodes giving the best combination. In parallel, GA provided the best clustering indexes, a similarity index of 0.599, Davies-Bouldin index lower than 0.112 and a sufficient value of dissimilarity index which confirms the significant distance between the formed clusters. Based on this study, various hazardous metal ions present in the tested water samples have been optimally classified. However, this system is more complex as it uses multiple electrodes, thus increasing the time and cost of fabrication [266]. Apart from HMs, genetic algorithm has been also used for the monitoring of other pollutants. Carreres-Prieto et al. have developed a spectrophotometry-based statistical model to estimate chemical oxygen

demand (COD), biological oxygen demand at five days (BOD$_5$), total suspended solids (TSS), phosphorus (P), total nitrogen (TN) and nitrate nitrogen (NO$_3^-$,N) of both raw and treated water without the need for any pre-treatment or chemicals. Multivariate linear regressions and machine learning genetic algorithm have been used to measure the spectral response of wastewater samples based on the absorbance and transmittance in the UVnear visible and visible 380–700 nm wavelength range. The obtained results have shown that the multilinear regression models can estimate only COD and TSS of raw water with less than a 0.5% error rate. In parallel, the models elaborated by means of GA can predict the degree of five pollutants (COD, BOD$_5$, TSS, TN and NO$_3^-$N) in both raw and treated wastewater with an error rate below 4% [264]. In contrast to the previously reported water quality assessment methods based on a single wavelength [267–270], the use of genetic algorithm, in this work was based on multiple wavelengths (81 wavelengths). Therefore, it improved the estimation accuracy of the pollution load of wastewater.

Other studies investigated the combination of Partial least squares (PLS) with genetic algorithm to extract pertinent information and produce truthful models. Nowadays, the PLS concept is applied by several researchers, according to Martens and Nea's algorithm [271,272]. However, the choice of wavelength would critically affect the future predictive ability of the model. To overcome this problem several selection methods have been developed, e.g., artificial neural network [273], Tabu search [274], hybrid linear analysis (HLA) [275], successive projection algorithm (SPA) [276], and genetic algorithm. In this context, a simple, fast and accurate procedure for quantifying malathion has been developed. Attenuated Total Reflectance-Fourier Transform Infrared (ATR-FTIR) spectra were used to select a specific region for the quantitative analysis using partial least square (PLS) and two wavelength selection methods: principal component regression (PCR) and genetic algorithm. Then, the relative error of prediction was calculated: 3.536, 1.656 for PLS and PCR-PLS, respectively, and 0.188 for GA-PLS, with correlation coefficient of 0.999. Based on this, we can say that GA can be considered as a helpful wavelength selection method for better capability of prediction [261]. Later, genetic-algorithm-based wavelength selection in multicomponent spectrophotometric determination by PLS was developed to detect a mixture of zinc and copper without a pretreatment. The method was based on the reaction between the analytes and a chromogenic reagent (2-carboxy2'-hyroxy-5'-sulfoformazylbenze (Zincon)) at pH 9. A series of synthetic solutions containing different concentrations of copper and zinc were used to check the prediction ability of the GA–PLS models. The root mean squares difference (the average error in the analysis) for copper and zinc with GA and without GA were 0.0407 and 0.0865, 0.2147 and 0.3005, respectively. Despite the high spectral overlapping observed between the absorption spectra of the mixture components, the obtained results proved the high potential of this approach to overcome the spectral interferences and detect simultaneously the cited ions in natural, tap and waste waters. Another pertinent point is that the selected variables clearly identify spectroscopically relevant regions, which demonstrated also the utility of GA for feature selection in a spectral data set [263].

In a recent study, Sajadi et al. used modeling and simulation of chemical reactions to construct a potentiometric AChE biosensor of Aflatoxin B1 by determining the optimal reaction rate constants. Enzymatic reactions were simulated using COMSOL software while reaction rates were optimized by ANN and GA. This method has shown satisfactory results with Mean Absolute Percentage Errors equal to 0.7074%, 0.9458%, 0.7473% and 0.7492% for train, validation, test and total data sets, respectively, and correlation coefficient higher than 0.999. As compared to the results obtained with other models used for predicting AChE enzyme inhibition by AFB1 [277,278], these results showed that trained Neural Network using Genetic Algorithm optimized reaction rates, exhibiting the lowest MAPE [224].

*4.4. Environmental Monitoring Based on Expert System (ES)*

In addition to genetic algorithm, expert systems are also used for environmental pollutants monitoring. An expert system devoted to voltametric methods for determining mercury (Hg), vanadium (V), and selenium (Se) has been developed. This work was an improvement on a previously described expert system for the determination of Cu, Zn, Cd, Pb, In, Ni, Co, Tl, Hg, V and Se, at trace levels [279,280]. The expert system was developed using a knowledge engineering system, in order to guide the user to choose the sample treatment and the most appropriate voltametric procedure for the identification and the quantification of the three trace metals. Four techniques were implemented: differential pulse polarography, anodic stripping voltammetry, cathodic stripping voltammetry and adsorptive stripping voltammetry, using mercury drop electrodes and a gold electrode. The authors demonstrated that the reported technique constitutes a very useful and reproducible approach, in particular for non-expert users in this field [281]. Based on ES, advanced mathematical techniques have also been designed in order to manage water quality monitoring networks. These studies include the development of a software tool for decision support, based on the application of fuzzy logic techniques and using the experience and knowledge of experts in this field. This system indicates water quality episodes from the behavior of variables measured at continuous automatic water control networks, which can hardly be detected by discrete sampling. Based on the recorded variables, the expert system provides different water quality phenomena indicators. These latter may be associated with a high probability of cause-effect relationship including human pressure on the water environment, urban discharges, or agricultural diffuse pollution. These indicators will forecast and complete manual sampling and laboratory analysis. Besides, this study demonstrated the ability of the expert system based fuzzy-logic to synthesize complex information, interpreted only by a few performant experts, and translated it into more understandable indicators for non-expert users [282].

In the literature, few reports have described ES in environmental monitoring, which can be explained by the necessity of deeper levels of understanding, interpretation, and expertise in the overall processes. The lack of knowledge available for a process leads to complex codification, in particular for activities that involve significant amounts of pattern recognition, generalization, and use of analogy [283]. Since an increasing number of researchers emphasize hybrid methods instead of single ones, more efforts should be focused on the use of ES combined with other strategies to involve contextual perspectives and develop high-quality models to monitor the environment [284,285].

*4.5. Environmental Monitoring Based on Computational Approaches*

In term of monitoring the environment, computational approaches have been explored to develop/redesign new bioreceptors or to reactivate biocatalysts for reactions of interest [152,162,286].

Computational approaches have been used to simulate a specific aptamer for diazinon, by studying molecular behavior and complex stability. The best sequence exhibiting a high affinity to bind Diazinon, was selected among twelve aptamers isolated from SELEX experimentation. Initially, Jokar's team used a docking technique as first virtual screening to select the most frequent conformation of each aptamer. Then, the docking results were used as inputs to molecular dynamics for a secondary screening. MD allows the simulation of the quantity and quality of aptamer–diazinon interaction, pointing out the conformational flexibility. In addition, it highlights the bonding interactions between diazinon–aptamer complexes showing the important role of binding energy in reinforcing this complex. Computational screening and analytical results showed that G-quadruplex DNA aptamer (DF20) with its stable complex, low oscillation, shifting toward folded states and high ratio of H-bond aptamer-Diazinon was the reliable candidate for diazinon biosensing. Based on simulation results, a colorimetric biosensing

platform was constructed for the ultrasensitive detection of Diazinon in the range of 0.141–0.65 nM with a LOD of 17.903 nM. Finally, the AuNPs-apta-sensing strategy was validated using a computational molecular approach. These computational approaches provide thus a promising alternative to laboratory experiments for receptor structure screening and the prediction of interaction outcomes [152].

Computational techniques also have a great potential application in studies redesigning the essential enzymes for OP hydrolysis and reactions that are not known to be catalyzed by natural enzymes. In this context a computational method has been developed to redesign a mononuclear zinc metalloenzyme for organophosphate hydrolysis. This approach aims to repurpose the reactivity of metalloenzyme active site functional groups for catalyzing new reactions. Based on this principle, Khareet al. engineered a zinc-containing mouse adenosine deaminase to catalyze the hydrolysis of an organophosphate substrate with a catalytic efficiency (*kcat*/*Km*) of ~$10^4$ M$^{-1}$·s$^{-1}$. In the high-resolution crystal structure of the enzyme, all the conception residues adopt the designed conformation except one residue. This computational enzyme design method could be considered as a general approach for exploring untapped catalytic potential for new reactivities [162]. Likewise, an optical biosensor for environmental monitoring has been developed based on protein modelingand computational screening followed by virtualmutagenesisanalyses. This approach was used for the engineering of functional amino acids in the D1 protein of the photosyntheticelectron transferchain of*Chlamydomonas reinhardtii*. These functions are able to detect two classes of pesticides, triazineandurea. The resulting protein was subsequently used to develop an optical biosensor for environmental monitoring with limits of detection between $0.8 \times 10^{-11}$ Mand $3.0 \times 10^{-9}$ M, depending on thetarget analyte (Figure 8) [287].

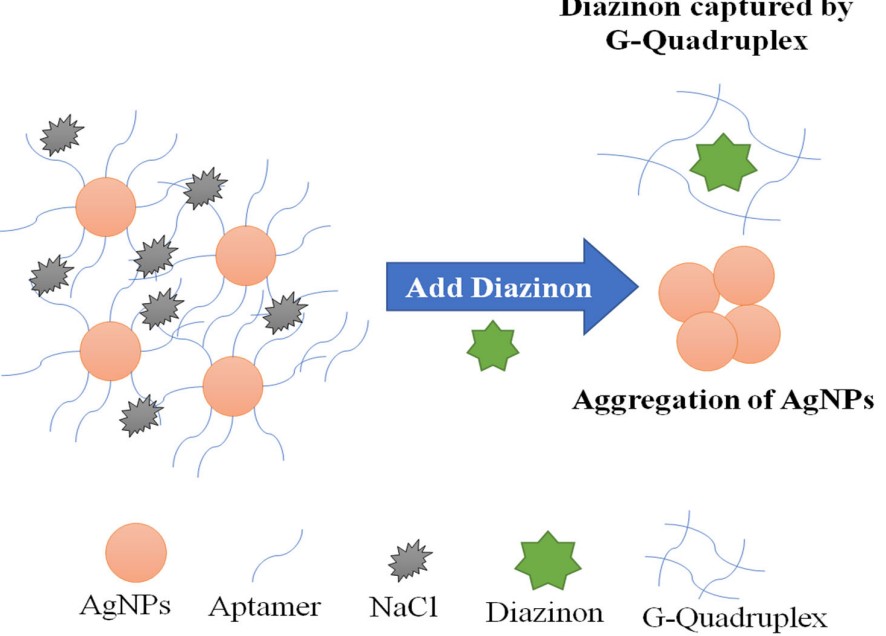

**Figure 8.** A schematic representation of the principle of Diazinon colorimetric aptasensing. Adapted from Jokar et al. [152].

Nanoparticles affect negatively living organisms(cytotoxic, neurotoxic, genotoxic, etc.) and environmental ecosystems, and in that sense computational methods have been used as an efficient tool to assess and evaluate the hazardous effects of toxic nanoparticles [288]. Because of the high-cost and time-consuming of the experiments, the quantitative

structure–activity relationship (QSAR) method is commonly used to predict the toxicity of different nanomaterials by developing nano-QSAR models [289].

Finally, the application of computational approaches in environmental pollution control is very limited in the literature. Further researches should be devoted to this AI technology in order to develop more reliable and robust devices to monitor pollutants that should not be underestimated [15,285].

## 5. Conclusions

Environmental pollutants such asheavy metals, pesticides, drugs, and biotoxins are extensively dangerous for all aspects of being organisms, including health, food, energy, etc. Monitoring systems traditionally used for these contaminants are limited by low efficiency, high cost and time consumption [290]. This paper reviewed the important applications and the recent progress in sensing strategies integrating artificial intelligence as a tool for modeling environmental monitoring. AI-biosensors are considered as an efficient analytical tool to detect the presence of one or more pollutants in complex samples with high sensitivity and selectivity. Furthermore, the ability of AI-approaches to learn by training and examples makes them flexible and powerful, and highly adapted for real time systems. In this context, numerous studies based on the integration of various AI technologies, including KBS, GA, and ANN, into numerical modeling systems have been discussed [22,254,291].

AI-biosensors in the environmental field are limited, in terms of the lack of applications in real samples comparing to medical applications. However, there are a few state-of-the-art biosensors for environmental monitoring, used for in situ operations and analytical performance. Therefore, there is a further challenge to develop improved and sensitive AI-tools to detect pollutants. Finally, AI-biosensing will provide a new platform for future innovation. In addition, considerable efforts are required to design reliable and robust devices that will enhance pollutant detection.

**Author Contributions:** J.-L.M.: supervisor; A.R. and A.T.: redaction. All authors have read and agreed to the published version of the manuscript.

**Funding:** This work received no external funding.

**Institutional Review Board Statement:** The study did not require ethical approval.

**Informed Consent Statement:** Not applicable.

**Data Availability Statement:** Data sharing not applicable.

**Conflicts of Interest:** The authors declare no conflict of interest.

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
