# Peer review of "Mathematical Modelling of Biosensing Platforms Applied for Environmental Monitoring"

_chemosensors, doi:10.3390/chemosensors9030050_

Round 1

Reviewer 1 Report

This is very well-written review of the role of mathematical tools for the biosensing of contaminants. This is a highly interesting issue that attracts increasing attention due to its potential to provide valuable information. The article is well organized and reviews a large number of references. Therefore, I recommend its publication in this journal since it will be of high interest for its readers.

Author Response

No remarks

Reviewer 2 Report

The considered manuscript is certainly an interesting and comprehensive review of the types of mathematical models applied up to date into an enviromental monitoring. Besides, it provides quite decent inventory of the substances most harmful environmentally. 
The are few omissions and suggestions:

  1. I've just read more detailed overview in the topic: Computational-nanotoxicology-challenges-and-perspectives , ISBN: 9789814800648 - I would suggest the Authors to get to know this book and quote it. (As well as the other papers by this group)
  2. The text of the manuscript is 'dense', and thus hard to follow - I would recommend to incorporate more schematic representations of the methods described here (as it was done for the ANNs here).
  3. As it was not clear for me, probably some explanation would be helpful, regarding the choice of the pollutants' listed in the Table 1 and 2 - why those and not the others? What was the criteria?

Author Response

1- I've just read more detailed overview in the topic: Computational-nanotoxicology-challenges-and- perspectives, ISBN: 9789814800648 - I would suggest the Authors to get to know this book and quote it. (As well as the other papers by this group)

Done. The passages are highlighted

2- The text of the manuscript is 'dense', and thus hard to follow - I would recommend to incorporate more schematic representations of the methods described here (as it was done for the ANNs here).

Two figures were added for the expert system and PCA (Principal Component Analysis).

3- As it was not clear for me, probably some explanation would be helpful, regarding the choice of the pollutants' listed in the Table 1 and 2 - why those and not the others? What was the criteria?

we opted for these pollutants because they are the most abundant substances in the environment, in addition they have the most hazardous effects not only on the ecosystem but also living organisms.

Reviewer 3 Report

The review presented by the authors is a useful summary of the state of the art. The work is well organised and documented. I support its publication.

Very minor issue: could you define LOD and LOQ ? I did not find that in the draft.

Author Response

1- Could you define LOD and LOQ? I did not find that in the draft

Done.